# Allogenic faecal microbiota transplantation for antibiotic-associated diarrhoea in critically ill patients (FEBATRICE)–Study protocol for a multi-centre randomised controlled trial (phase II)

Ivana Cibulkova[1,2], Veronika Rehorova[2,3], Hana Soukupova[2,4], Petr Waldauf[2,3], Monika Cahova [5], Jan Manak[6], Martin Matejovic[7], Frantisek Duska[2,3]*

1 Division of Gastroenterology, Department of Internal Medicine, Kralovske Vinohrady University Hospital, Prague, Czech Republic, 2 The Third Faculty of Medicine, Charles University, Prague, Czech Republic, 3 Department of Anaesthesia and Intensive Care Medicine, Kralovske Vinohrady University Hospital, Prague, Czech Republic, 4 Department of Microbiology, Kralovske Vinohrady University Hospital, Prague, Czech Republic, 5 Department of Experimental Medicine, Institute for Clinical and Experimental Medicine, Prague, Czech Republic, 6 3rd Department of Internal Medicine–Metabolism and Gerontology, Charles University Teaching Hospital Hradec Kralove, Hradec Kralove, Czech Republic, 7 1st Department of Internal Medicine, Faculty of Medicine in Pilsen, Pilsen University Hospital, Pilsen, Czech Republic

* frantisek.duska@lf3.cuni.cz

## Abstract

### Background

Exposure of critically ill patients to antibiotics lead to intestinal dysbiosis, which often manifests as antibiotic-associated diarrhoea. Faecal microbiota transplantation restores gut microbiota and may lead to faster resolution of diarrhoea.

### Methods

Into this prospective, multi-centre, randomized controlled trial we will enrol 36 critically ill patients with antibiotic-associated diarrhoea. We will exclude patients with ongoing sepsis, need of systemic antibiotics, or those after recent bowel surgery or any other reason that prevents the FMT. Randomisation will be in 1:1 ratio. Patients in the control group will receive standard treatment based on oral diosmectite. In the intervention group, patients will receive, in addition to the standard of care, faecal microbiota transplantation via rectal tube, in the form of a preparation mixed from 7 thawed aliquots (50 mL) made from fresh stool of 7 healthy unrelated donors and quarantined deep frozen for 3 to 12 months. Primary outcome is treatment failure defined as intervention not delivered or diarrhoea persisting at day 7 after randomisation. Secondary outcomes include safety measures such as systemic inflammatory response, adverse events, and also diarrhoea recurrence within 28 days. Exploratory outcomes focus on gut barrier function and composition of intestinal microbiota.

relevant data from this study will be made available upon study completion.

**Funding:** Donatio Intensivistam Endowment Fund.

**Competing interests:** The authors have declared that no competing interests exist.

**Abbreviations:** AAD, antibiotic-associated; ICU, Intensive Care Unit; HDU -, High Dependency Unit; C. dif, *Clostridioides difficile*; eCRF, Electronic Case Report Form; FMT, Faecal Microbiota Transplantation; SUKL, The State Institute for Drug Control; DMSB, Data Monitoring and Safety Board; PI, Principal Investigator; NEC–, Necrotizing enterocolitis.

## Discussion

Faecal microbiota transplantation has been effective for dysbiosis in non-critically ill patients with recurrent *C. difficile* infections and it is plausible to hypothesize that it will be equally effective for symptoms of dysbiosis in the critically ill patients. In addition, animal experiments and observational data suggest other benefits such as reduced colonization with multi-drug resistant bacteria and improved gut barrier and immune function. The frozen faeces from unrelated donors are immediately available when needed, unlike those from the relatives, who require lengthy investigation. Using multiple donors maximises graft microbiota diversity. Nonetheless, in vulnerable critically ill patients, Faecal microbiota transplantation might lead to bacterial translocation and unforeseen complications. From growing number of case series it is clear that its off label use in the critically ill patients is increasing and that there is a burning need to objectively assess its efficacy and safety, which this trial aims.

## Trial registration

www.clinicaltrials.gov (NCT05430269).

## Background and rationale

All eukaryotic organisms, including humans maintain a close relationship with their microbiota, which comprises bacteria, archaea, viruses, and fungi. Dysbiosis, an imbalance in the composition and/or diversity of these microorganisms, is linked to various diseases [1, 2]. The human microbiota composition is individual-specific, influenced by factors such as birth mode, diet, genetic traits, host immune status, infections, and drug use [3, 4]. In intensive care units, 70% of patients are administered antibiotics [5, 6], frequently leading to dysbiosis. This can present as diarrhoea or, more severely, as *Clostridioides difficile* infection (CDI) [5, 7, 8]. A particular dysbiosis pattern appears prevalent in critically ill patients, marked by an increase in Enterobacter and Staphylococcus species and a decrease in symbiotic Firmicutes or Faecalibacteria [6, 9–11]. Resulting diarrhoea can cause skin irritation, odour, pain, and loss of dignity. Additionally, it increases the risks of ion imbalance, malnutrition, and overall morbidity and mortality [12, 13]. Interventions for antibiotic/associated diarrhoea in ICU currently include prebiotics, probiotics, and Faecal microbiota transplantation (FMT). However, none have robust evidence backing them [14–16]. Using prebiotics in fibre-containing enteral nutrition formulas is debated due to bowel ischaemia risks and unproven effects. Although there has been a substantial number of studies examining probiotics in various gastrointestinal diseases, the studies have been extremely varied. According to available data, probiotic treatment (with defined strains) is recommended only in patients with pouchitis or in preterm infants to prevent the development of NEC. While for other indications such as CDI prevention, antibiotic associated diarhea, IBD therapy, the data are not uniform and their use cannot be clearly recommended and should be limited to the context of a clinical trial [16]. Similary, their benefit for patient-centered outcomes in the critically ill remains unproven [17–19]. Moreover, they might harm specific subgroups, like patients with severe pancreatitis [20, 21]. Therefore, current standard of treatment for antibiotic-associated diarrhoea includes the avoidance of further antibiotic exposure, when possible, hydration with i.v. fluids and correction of electrolyte and

acid base abnormalities. In addition, natural purified clays such as diosmectite are used to absorb bacterial toxins and are also believed to protect gastrointestinal mucosal layer.

Introduced in the 1960s, FMT directly impacts gut microbiota [22, 23].It involves transferring minimally processed faeces from a healthy donor to a recipient's digestive tract. This technique is established as effective for recurrent CDI [2, 24, 25]. Some case studies suggest FMT's ease and safety in ICU patients with recurrent CDI and severe colitis, where colectomy is considered [8, 26–36]. Animal models hint at FMT's potential to clear multidrug-resistant organisms from the patient's intestine [29, 30]. However, most available information on FMT effects in critically ill patients comes from uncontrolled studies. The number of published case reports and case series considered [2, 26–28, 31, 34–36] is increasing over time, suggesting increasing off-label and experimental use. The fact, that the effects of FMT are unequivocally positive and without complication cannot be considered as evidence of its efficacy and safety due to publication bias. Indeed, critical illness makes patients also prone to developing complications of FMT. Gut mucosal barrier is impaired and a degree of bacterial translocation during FMT combined with dysregulated immune system in the critically ill patients put the safety of the procedure in this population in question. Only controlled studies can bring the answers.

In the light of this we designed a randomized prospective controlled trial to assess in the critically ill patients with antibiotic-associated diarrhoea the effects of FMT compared with standard of care on the hazard ratio persistent diarrhoea after 7 days of treatment and a range of secondary safety outcomes.

## Methods

### Trial aim and hypothesis

We hypothesise that FBT will be feasible as first-line treatment of postantibiotic diarrhoea in the critically ill, including those with uncomplicated CDI, and that the intervention will be superior to the standard in the ability to reduce the incidence of diarrhoea on day 7 after randomisation.

### Trial design

This study is a prospective, randomized (1:1), open-label, assessor-blinded, controlled, outcome-based superiority, trial with two parallel groups across multiple centres. The study was conducted according to the SPIRIT reporting guidelines (S5 Appendix) [37].

### Study setting

Intensive care units of university hospitals in the Czech republic, who treat unselected adult critically ill patients. This includes the coordinating centre FNKV University Hospital, a tertiary 2000-bed teaching hospital in Prague, Czech Republic and collaborating institutions such as VFN General University Hospital, Hradec Kralove University Hospital (FNHK) and Plzeň University Hospital (FNP). All the participating centers routinely use Bristol scale to assess stool consistency and have a robust and detailed system of documenting frequency bowel movements in all patients.

### Eligibility criteria

Patients eligible for the trial must fulfil all the following inclusion criteria:

- Age > 18 yrs

- In-patient in ICU or HDU (including burn unit) and expected to stay for >7 days.

- Diarrhoea following antibiotic treatment defined as 3 or more stools per day or Bristol type 7 stool in the volume >300 ml/day if stool derivative device is in place, persisting for at least 24 hours

- Systemic antibiotics had been administered during last 2 weeks but are stopped >24 hours ago

- Clinical team committed to full active treatment

- Written prospective informed consent

     None of the following <u>exclusion criteria</u> must be present for patient to be eligible for enrolment:

- Death appears imminent

- Active sepsis defined as per 2016 definition

- Lactate >2.0 mM or clinical signs of haemodynamic instability, which includes but is not limited to vasopressor requirements ≥0.2 ug/kg.min

- Colon diameter > 9 cm on plain AXR or any clinical suspicion of toxic megacolon

- Necessity of ongoing antibiotic treatment for another reasons.

- Unable to tolerate enema for any reason (e.g. recent surgery of the GI tract)

- Pregnant and lactating woman

- Patients with a history of severe anaphylactic food allergy

- Neutropenia (neutrophil count below $1.0 \times 10^9$/L)

- Graft versus host disease

- Short bowel syndrome, Malabsorption syndrome or suspicion of osmotic diarrhoea

- Infectious cause of diarrhoea

- Any other reason which–as per judgement of the investigator–makes faecal transplantation unsafe or not feasible (Note: All screening failures based on this criterion will be reported separately, including the reason why it was considered unsafe or not feasible to proceed).

### Interventions

**Intervention group.**   After obtaining baseline data (see below), in patients randomized to intervention, Faecal microbiota transplantation is performed as soon as possible but always within 24 hours after enrolment. Standard of care for antibiotic-associated diarrhoea is pre-scribed or continued.

*The FMT procedure*. For details FMT mixture preparation including donor examination, see Supplementary data file (S1 Appendix). Two hours before the procedure, the patient receives 2 mg of oral loperamide. Enteral nutrition or oral intake is discontinued 1 hour before loperamide administration. The patient is positioned into left semi-lateral Trendelenburg posi-tion and 350 ml of prewarmed faecal mixture is inserted via a soft rectal tube inserted as far as possible into the colon. The patients remain in the position for the next 15 minutes after which they are repositioned to right semi-lateral Trendelenburg position for the next 15 mins. Most patients will have faecal collection system in place, or it is inserted after the FMT

administration. The outlet tube is cross clamped for one hour whilst the patient is closely observed. In case of abdominal pain or spasms, the faecal collection system is immediately open. In view of efforts to reduce peristalsis to maintain transplant in the colon, oral or enteral nutrition can be resumed no earlier than 2 hours after FMT.

In the subset of patient who are known to be *Clostridioides difficile* positive, the procedure is modified following way: Three hours before FMT, colon is irrigated with 1L of pre-warmed normal saline, which is then allowed to drain freely into the faecal derivation system. When most of the instilled solution is drained, 2 mg of loperamide is given orally and FMT procedure is performed 2 hours later. Any clinical worsening, ne onset of sepsis or if diarrhoea is not improving by four, enteral vancomycin (125mg every 6 hours) will be started and the patient would be deemed as treatment failure regardless of regardless of clinical cure status at day 7.

Intervention adherence and protocol implementation. If the patient condition changes at any time between randomisation and FMT procedure, the treating clinician or investigator may decide not to perform FMT procedure whenever they deem that risk outweigh benefits. In addition, patients may decide to withdraw consent or the FMT procedure turns out not feasible or not delivered for any reason, e.g. technical or logistical. The occurrence of any of these conditions will mean that the patient is considered treatment failure in the intention-to-treat analysis, regardless of clinical cure status at day 7. In case of clinical progression, therapy will be escalated according to current recommendations.

**Control group.**   In patient randomized to the control group, after obtaining baseline dataset, the standard of care for antibiotic-associated diarrhoea will be prescribed or continued. This includes enteral diosmectite 2g every 8 hours for 5 days, parenteral or enteral rehydration, correction of acid/base and electrolyte abnormalities, guided by regular clinical and laboratory monitoring as per discretion of the treating clinicians.

This treatment will be modified in the subgroup of patient with *Clostridioides difficile* infection by adding oral vancomycine (125mg every 6 hours). In case of clinical progression, therapy will be escalated according to current recommendations.

## Concomitant care

No probiotics shall be administered to patients enrolled into the trial. Pharmaconutrition including fibre-containing enteral nutrition formulas are permitted provided these are considered as the standard of practice in the participating centre and consistent across the treatment arms, but no modifications of enteral nutrition strategies aimed at influencing bowel motility should be performed for patients in the trial. Enteral opioids (e.g. loperamide) should be avoided with the exception of the single dose to enhance graft retention after the FMT, as described above. All other aspects of concomitant care are left to the discretion of treating clinicians. This includes the administration of systemic antibiotics, which should generally be avoided in antibiotic-associated diarrhoea. If the clinical condition necessitates systemic antibiotics, patient will remain in the trial and all outcomes will be reported in intention-to-treat population, with the exception of exploratory analyses, which will be performed on per-protocol basis in patients who did not receive systemic antibiotics within the first 7 days into the trial.

## Outcomes

### Primary outcome.

- Hazard ratio of treatment failure in the intervention group related to the control group at day 7 after randomization, which is defined as treatment either had not been delivered or the

diarrhoea persists (based the number of bowel movements and assessment of stool consistency as per Bristol scale), it the intention-to-treat population.

**Secondary outcomes.** All the analyses of secondary outcome will be performed in the intention-to-treat population.

- Relative risk ratio of composite adverse events, consisting of such as new-onset sepsis, toxic megacolon, positive blood culture, or any serious adverse event suspected from being related to study intervention.

- SOFA score at days 4 and 7

- Odds ratio of being alive and relapse-of-diarrhoea-free at the time of hospital discharge or on day 28, whichever occurs first.

- In-hospital mortality at day 28

- Hazard ratio of treatment failure in the intervention group related to the control group at day 7 after randomization, which is defined as treatment either had not been delivered or the diarrhoea persists, in the subgroup of patients with *Clostridioides difficile*.

- Odds ratio of being alive and relapse-of-diarrhoea-free at the time of hospital discharge or on day 28, whichever occurs first, in the subgroup of patients with *Clostridioides difficile*.

## Exploratory outcomes

Analyses of exploratory outcomes will be performed in the per protocol population excluding those who received systemic antibiotics within first 7 days into the trial. The objectives are:

- To determine the influence of the intervention on colonic microbiome (diversity, total bacteria abundance and composition of fecal microbiome) by quantitative sequence-based mapping

- To assess bowel barrier integrity, bacterial translocation and inflammation by plasma markers (citrulline, IL-6, IL-8, IL-12, TNF-a, LL-37, cathelidicine, fatty acid binding proteins, plasma LPS), fecal markers (calprotectine)

- To assess the donor's fecal microbiome diversity and viability by quantitative sequence-based mapping, cultivation and flow cytometry

- To determine whether and exactly which bacteria from the donor stool have become established in the recipient's intestine after FMT and to find any associations with the clinical outcome

- To compare inflammatory response between intervention and control groups during the first 7 days in the study using area under the curve of SIRS score or biomarkers such as CRP.

## Participant timeline

Study procedures are summarised in Fig 1.

The clinical symptoms of diarrhoea and signs of possible complication of diarrhoea and FMT are the main subject of interest in this study. The patients will have study visits daily as long as they stay on ICU or until day 7. Primary outcome will be recorded at day 7 after enrolment. At discharge from ICU (which may occur before or after day 28) a summary of adverse

| TIMEPOINT | Enrolment | Allocation D0 | D1 | D2 | D3 | D4 | D5 | D6 | D7 | D14, 21, 28 | Close-out 6month follow up |
|---|---|---|---|---|---|---|---|---|---|---|---|
| **STUDY PERIOD** | | | | | | | | | | | |
| | Enrolment | Allocation | | | | Post-allocation | | | | | Close-out |
| **ENROLMENT:** | | | | | | | | | | | |
| Eligibility screen | x | | | | | | | | | | |
| Informed consent | | x | | | | | | | | | |
| Pregnancy test (in fertile women) | | x | | | | | | | | | |
| Stop diarrhoea contributing medication | | x | | | | | | | | | |
| Abdominal X ray | | x | | | | | | | | | |
| Allocation | | x | | | | | | | | | |
| **INTERVENTIONS:** | | | | | | | | | | | |
| *Intervention group:* | | | | | | | | | | | |
| Loperamide 2 mg p.o. | | | x | | | | | | | | |
| FMT retention enema | | | x | | | | | | | | |
| Symptomatic therapy (hydration, Diosmectite 2g every 8 hours for 5 days) | | | ◆———————————◆ | | | | | | | | |
| **Subgroup CD positive patients** | | | | | | | | | | | |
| 1 L saline enema | | | x | | | | | | | | |
| FMT retention enema | | | x | | | | | | | | |
| *Standard care group* | | | | | | | | | | | |
| Symptomatic therapy (hydration, Diosmectite 2g every 8 hours for 5 days) | | | ◆———————————◆ | | | | | | | | |
| **Subgroup CD positive patients** | | | | | | | | | | | |
| Vancomycin 125mg p.o. á 6hrs, 10days | | | ◆———————————————————◆ | | | | | | | | |
| **ASSESSMENTS:** | | | | | | | | | | | |
| Recording the stool frequency and Bristol Scale | | x | x | x | x | x | x | x | x | x | x |
| Adverse events | | x | x | x | x | x | x | x | x | x | x |
| Recording the ATB treatment | | x | x | x | x | x | x | x | x | x | x |
| Phys.exam, SOFA | | x | x | x | x | x | x | x | x | x | x |
| Serum sample analyze + freeze | | | x | | | x | | x | x | x | x |
| Stool sample analyze + freeze | | x | x | | | x | | | x | x | x |
| Blood culture | | | x | | | x | | | | | |

Note: C.dif. = *Clostridioides difficile* infection; D/C = discharge from ICU.

**Fig 1. SPIRIT schedule of enrollment.** Note: C.dif. = *Clostridioides difficile* infection; D/C = discharge from ICU.

events, stool characteristics and antibiotic treatment will be recorded. The expected duration of participation in the study is 6 months. The last follow up visit will be performed in and outpatient clinic or, if the patient is not able or willing to come, over telephone interview. All data will be recorded in a custom-made electronic Case Report Form.

## Sample size calculation

There is no outcome data in critically ill adults with AAD. However, in two trials in non-critically ill patients, the success of FMT in primary Clostridioides infection was 69% (22 out of 32) or 56% (5 out of 9) as compared to 45% cured by oral metronidazole [38, 39]. Assuming that clinically significant difference between the two methods is 20% and the FMT will have identical efficacy in stopping the AAD in ICU patients, 36 patients are required to have 80% chance that the upper limit of a one-sided 95% confidence interval (or equivalently a 90% two-sided confidence interval) will exclude a difference in favour of the standard group of more than 20%.

## Recruitment

Each center will screen subjects separately in a competitive fashion until the target sample size is achieved (36 subjects). A dedicated research nurse or investigator will identify eligible patients every day during morning rounds and all eligible patients, or their representatives will be approached and offered participation in the study. With 70% of ICU patients being exposed to antibiotics [5, 6], out of which 1 in 5 develops AAD [40], out of which approximately one half stay on ICU for more than a week, we assume 1 enrolled subject per 50–100 patients admitted to ICU, depending on consent rate, which is currently unknown.

## Randomization

Participants will be randomly assigned treatment allocation using computer-generated random sequence of numbers using random sequence script in software R (http://www.randomization.com), which is embedded in electronic case report form (eCRF). Randomization code is released only after baseline data have been entered and the patient has been recruited into the trial. Randomization is stratified according to treatment center and the presence or absence of known C. dif. Positivity at the time of randomisation. The block sizes is not disclosed to ensure concealment.

## Blinding

This is an unblinded study and the investigators and clinical team will be aware of patient treatment allocation. When assessing primary outcome, an independent outcome assessor unaware of patient treatment allocation will extract outcome data from nursing notes. Emergency unblinding procedure is not applicable.

## Data collection plan

Before the study initiation in each center, personnel will be trained by the central research coordinator on study processes, documentation, and the use of the e-CRF system. The eCRF is designed for feasible data entry during staff shortages and high workloads.

For primary outcomes, stool quantity and consistency will be documented using the Bristol scale [41] on day 7 within the e-CRF. An independent analyst will review nursing notes from two 12-hour shifts ending on day 7 to record diarrhoea presence or absence. If the patient is outside the ICU but within hospital care on day 6, they will be given a 24-hour stool data

collection form. Patients discharged will be contacted a day ahead and asked to provide the following day's stool sample and form either to the hospital or via a research staff's home visit.

Additional metrics include the SOFA score, per international guidelines. Data on events like toxic megacolon, positive blood cultures, or other severe events possibly related to FMT will be recorded. Daily monitoring will cover adverse events, antibiotic treatments, and stool details for ICU patients using validated Bristol scale [41]. Follow-up visits, similar to Day 4 procedures, will be conducted every 7 days until day 28 or ICU discharge. Details on individual filtrate aliquots for FMT, including donor specifics, test results, processing methods, and storage data, will be logged. The results of the FMT mixture examination will also be recorded. Non-critical data may be entered retrospectively.

## Retention strategy

After patient randomization, each study site will ensure continued tracking of the participant. Primary outcome of this study will be collected whilst the patient is still in ICU, early unanticipated deaths are rare in stable patients in ICU and patients discharged from ICU before day 7 will be pro-actively visited by a research nurse. We therefore we aim to achieve complete dataset for primary outcome. Scheduled follow-ups on days 14, 21, 28, and at 6 months will take place either as inpatient visits, outpatient appointments, or phone calls. If patients are in the hospital but outside the ICU, a research nurse will see them on days 7, 14, 21, and 28. Out-of-hospital patients will be asked to come to the outpatient clinic for a physical check and blood test, and if possible, to provide a stool sample. For those unable to attend in person, a phone interview will be conducted. We anticipate a loss to follow-up rate of no more than 30% after hospital discharge, mostly due to consent withdrawals or post-ICU deaths.

## Data management

Data will be entered into a custom-made electronic Case Report Form (eCRF), accessible online, ensuring ease of input and consistent data capture across sites. To enhance data quality, range checks will be implemented to identify any out-of-range or inconsistent data values. All identifiable patient data will be stored on a secure hospital server. After the study's completion and following a six-month embargo period, deidentified datasets will be made available for secondary research using the REDCap (Research Electronic Data Capture) platform, a secure, web-based application designed to support data capture for research studies. Deidentification will involve removing or anonymizing all personal identifiers, including names, addresses, and other direct identifiers, as well as employing pseudonymization techniques for indirect identifiers. We will also use REDCap tools for access control and establish data-sharing agreements. The study's data management practices will comply with ethical guidelines and legal requirements, including GDPR, ensuring responsible data stewardship.

## Statistics

For primary outcome we will calculate unadjusted hazard ratio (with 95% interval) of not being cured in intention-to-treat population.

Same analysis will be performed for the subgroup of patients with CDI as secondary outcome. Odds ratios with 95% confidence intervals of being alive and free of diarrhoea at day 28 or hospital discharge, whichever occurs first, will be calculated for whole population and for subgroup of patients with CDI. SOFA scores will be compared by methods of descriptive statistics.

The microbiome composition and diversity (exploratory outcomes) will be assessed using available R software packages and in-house tools. For individual tasks, the following R

packages will be used: DESeq2 [42], vegan (PERMANOVA), effsize (Cliff's delta), glmnet (Lasso/Ridge regression), and caret (machine learning library). DESeq2 normalization method is used to normalize microbiome data to account for differences in sequencing depth, which is crucial for ensuring that comparisons of microbial abundance are not biased by variations in sample sequencing. It provides variance-stabilized transformation, which is particularly useful for comparing differential abundance across groups. Vegan PERMANOVA method is chosen for its ability to assess differences in microbial community composition between groups. PERMANOVA (Permutational Multivariate Analysis of Variance) is a non-parametric method that can handle the complexity and high-dimensional nature of microbiome data, allowing us to determine whether there are significant compositional differences in the gut microbiota of patients before and after FMT. Lasso/Ridge Regression with glmnet techniques are applied to identify and quantify associations between specific microbial taxa and clinical outcomes. Lasso regression helps in feature selection, potentially identifying the most relevant bacteria linked to patient outcomes, while Ridge regression stabilizes estimates then predictors are highly collinear, a common issue in microbiome studies.

The microbiome data will be treated as compositional (proportions of total read count in each sample, nonrarefied), and prior to all statistical analyses will be transformed using DESeq2 normalization by variance Stabilizing Transformation function. Only the diversity indexes will be calculated on rarefied data. No imputation of missing data will be made. We will record and report reasons for withdrawal for each randomization group and compare the reasons qualitatively.

## Data monitoring

Data monitoring safety board (DMSB) consists of established senior specialists in the field of intensive care medicine and infectious diseases and a nurse representative. All members must be independent on the study team and unrelated to funding bodies. DMSB will review any SUSAR and has the right to terminate the trial at any points.

## Interim analyses

After the primary outcome of the eight subjects enrolled to the trial is known and e-CRF is completed, DMSB will review–in an unblinded fashion–the records of all enrolled patients, focusing on the occurrence of adverse events and safety of FMT in the critically ill patients. No formal statistical analyses will be performed at this stage.

After the primary outcome of 20[th] subject is known and e-CRF is completed, the study statistician will prepare (a) cumulative report of adverse events and (b) interim analysis of primary outcome with post-hoc power analysis and presents it to DMSB. The DMSB may stop the trial for safety reason or, based on interim analysis issue recommendation to either stop the trial for continue as planned or, alter the target sample size based of post-hoc power analysis.

## Harms and safety measures

While FMT offers potential benefits and has demonstrated safety in ICU patients [6, 43], limited data exist regarding its use in individuals with potentially compromised intestinal barriers and low quality reports are subjected to publication bias. To ensure utmost safety in our study, several measures are taken: Clinicians administering treatments will remain unblinded, proactively addressing any emergent safety concerns. Vital parameters such as signs, blood cultures, endotoxin levels, and early cytokine measures will be rigorously assessed both before and three hours post-FMT, targeting insights into potential bacterial translocation. Standard protocols dictate continuous monitoring of all vital functions of patients. Additionally, stool

characteristics, including frequency and volume, will be documented every 12 hours, guided by the Bristol scale. Specific clinical thresholds, such as sepsis onset, lactate levels surpassing 2mM, or a marked rise in inflammatory markers accompanying pyrexia above 38°C without an evident alternative cause, will necessitate the administration of targeted intravenous antibiotics. The introduction of such antibiotics, particularly those effective against C. diff., will be at the clinician's discretion, with the rationale for their use meticulously recorded.

Our stringent safety monitoring includes a prompt review of any Suspected Unexpected Serious Adverse Reaction (SUSAR) by the DMSB within 48 hours of its occurrence. The emergence of two SUSARs in the intervention group warrants an immediate halt to the trial. Moreover, an expert team will convene on an as-needed basis to address arising safety concerns, in addition to the pre-planned safety interim analyses as described above.

## Auditing

The trial may be subjected to audits by regulatory authorities, specifically The State Institute for Drug Control (SUKL). Monitors will evaluate source documents, which include medical charts, e-CRFs, initial hospital admission reports, and associated records, ensuring their completeness and accuracy. Critical variables such as patient initials, date of birth, gender, informed consent, eligibility criteria, randomization date, treatment specifics, adverse events, and endpoints will be verified for all participants. Site investigators will also ensure all pertinent documentation is current. Should discrepancies or issues be detected, monitors will assist site personnel in rectifying the identified concerns.

## Research ethics approval

The Ethics Committee for Multi-Centric Clinical Trials (EC) of FNKV University Hospital approved the trial on 2.6.2021, decision reference No KH/40/00/2021. The conduct of the study including informed consent procedure is compliant with the latest 2013 Amendment of the Declaration of Helsinki (ver. 2013). In addition, local research ethics board of any participating hospital will review and approve the protocol. Any changes to the protocol will be submitted for research ethics board review prior to implementation.

Patients with decision-making capacity potentially meeting the eligibility criteria will be approached and offered participation in the study. They will provide written prospective informed consent and consent for the use and storage of personal data as per General Data Protection Regulation (GDPR). It is expected that a significant proportion of patients who meet the entry criteria will not have the capacity to provide informed consent due to the condition for which they were admitted to the ICU, such as unconsciousness or the influence of sedatives. To these patients a deferred consent policy will be applied, which means that prior to enrollment, patients' legal representative will be asked to give consent on their behalf. The legal representative will be usually the next of kin, or for those whose relatives are not available despite reasonable effort, an independent physician. As soon as patients regain the capacity, they will be informed about their enrolment into the study and offered to continue into the study, to withdraw with permission to use the data collected up to the point or to withdraw from the study and request deleting all data collected. There is only one version of informed consent form, no separate consent for ancillary studies will be sought.

## Confidentiality

All information related to the study will be securely kept at the study site, with restricted access. Any laboratory results, reports, and forms will only display a coded ID number to ensure participant confidentiality. All data will reside in a password-secured database

connected to the eCRF, and safeguarded on the hospital's secure servers. Entry of primary data into the eCRF will be done either by the investigators or specialized study staff, always under investigator oversight. The responsibility for the data's quality and integrity lies with the investigators.

## Ancillary and post-trial care

Sponsors and investigators are committed to addressing any health concerns directly linked to participation in the study. All study participants are ensured protection under specialized insurance provisions tailored for trial subjects. Post-trial, participants will continue to receive standard care without any modifications due to their involvement in the study, and this care will be provided by teams not affiliated with the trial itself.

## Dissemination policy

**Trial results.**   The investigators and sponsor prioritize transparent communication of the trial's findings to ensure the informed engagement of participants, healthcare professionals, the public, and other key stakeholders. Participants will receive a comprehensible summary of results, while healthcare professionals will be updated through presentations at relevant conferences and publications in peer-reviewed journals. The general public will be informed via press releases and, when feasible, through summaries on the trial's official website. Additionally, after a 6-month embargo following the publication of primary outcomes, de-identified patient data will be publicized for secondary analyses. All publications will undergo a review by the investigators and the sponsor to ensure clarity, compliance, and preservation of the study's integrity.

**Authorship.**   In accordance with the International Committee of Medical Journal Editors' criteria for authorship, all potential authors must have significantly contributed to study conception and design, data acquisition or interpretation, manuscript drafting or revising for intellectual content, and have approved the final version for publication. FD, as the Chief Investigator, initiated the study, handled site selection and contracts, and led both proposal and protocol formulation. IC, VR, and HS are physicians overseeing protocol compliance and managing FMT methodology. PW manages study statistics and data analysis, while MC, a microbiology specialist, focuses on analyzing and interpreting microbial shifts. PB acts as the independent safety advisor. Each author played a crucial role in shaping the study design and proposal, and all have reviewed and approved the final manuscript.

**Reproducible research.**   De-identified participant-level data- set will be made available 6 months after the publication of the results of the study upon reasonable request to the principle investigator.

## Discussion

Faecal microbiota transplantation has been effective for dysbiosis in non-critically ill patients with recurrent *C. difficile* infections [2, 24, 25] and it is plausible to hypothesize that it will be equally effective for symptoms of dysbiosis in the critically ill patients. In addition, animal experiments [29, 30] and observational data [10, 44–48] suggest other benefits such as reduced colonization with multi-drug resistant bacteria and improved gut barrier and immune function. From growing number of case reports and case series [26–28, 49–53] it is obvious that FMT is being increasingly used off label in the critically ill patients, including during life-threatening conditions such as toxic megacolon [26, 28, 54–56]. Indeed FMT might be a risky procedure in ICU patients who may have gut barrier function disrupted, e.g. by non-occlusive ischaemia during circulatory shock or by changes of luminal microbiota. Therefore, we there

is a burning need to test FMT during strictly controlled conditions first in the population of critically ill, who are haemodynamically stable and free of sepsis at the time of enrolment. Only later clinical trials shall be expanded to more unstable ICU patients. This is particularly important for the subset of patients with C. dif. infections.

The choice of unrelated healthy donors was pragmatic. The frozen faeces from unrelated donors are immediately available when needed, unlike those from the relatives, who are not always willing to donate, and even if they do, their investigation takes time. Quarantine of the sample reduces the risk of transmission of contagious diseases as the stool is only used after second set of tests in the same donor 3 months later. This allowed us to choose multi-donor approach, which hinges on the notion that an ideal donor's microbiome should closely mirror the recipient's premorbid microbiome [57]. Recognizing the inherent microbial diversity among healthy individuals, we hypothesized that exposing the ICU patient's intestinal mucosa to a broader array of microbial species (via multidonor sources) could be advantageous. This multidonor approach aims to offer the recipient's gut a vast spectrum of bacterial species, potentially enabling colonization closer to its natural, pre-disease state [57–59].

Dose selection is an intricate consideration given the current literature does not clearly recommend the optimal number of FMT administrations. In this study prioritizing safety, a single FMT dose was chosen for patients in the intervention arm, with a second dose only for those with a confirmed CD infection. The practical choice of seven 50 mL aliquots, totalling 350 mL, is a compromise between volumes documented in previous research: 500 mL showing 97% success and 200 mL achieving 80% success [57, 60, 61]. Lastly, leveraging insights from past studies, we determined that delivering FMT through enema would be the most efficacious and safest route, also aligning with implementation simplicity [25, 60, 61].

In conclusion, by integrating robust scientific approaches with pragmatic considerations prioritising safety, this study aims to provide the first controlled evaluation of FMT's potential in critically ill patients.

## Trial status

Overall Status: Recruiting
 Record Verification: February 2023
 Study Start: February 9, 2023 [Actual] Primary Completion: December 1, 2024 [Anticipated]
 Study Completion: July 1, 2025 [Anticipated]

## Supporting information

**S1 Appendix. Donor management and faecal microbiota transplant preparation and storage.**
(DOCX)

**S2 Appendix. Donor questionnaire.**
(DOCX)

**S3 Appendix. Biological specimen.**
(DOCX)

**S4 Appendix. Flowchart.**
(TIF)

**S5 Appendix. Spirit checklist.**
(DOCX)

**S1 File.**
(DOCX)

## Acknowledgments

We would like to thank all the volunteers who decided to participate in this study and research assistants, namely Šárka Gregorová, Nikola Bandíková, Šárka Vosalová, Kateřina Ťopková and Marie Chaloupecká for their valuable contributions to this project. The authors also thank Irena Odstrcilova for her help with study administration.

## Author Contributions

**Conceptualization:** Ivana Cibulkova, Veronika Rehorova, Monika Cahova, Martin Matejovic, Frantisek Duska.

**Data curation:** Petr Waldauf, Monika Cahova.

**Formal analysis:** Petr Waldauf, Monika Cahova.

**Funding acquisition:** Frantisek Duska.

**Investigation:** Veronika Rehorova, Hana Soukupova, Jan Manak, Martin Matejovic.

**Methodology:** Ivana Cibulkova, Hana Soukupova, Frantisek Duska.

**Project administration:** Veronika Rehorova.

**Visualization:** Veronika Rehorova.

**Writing – original draft:** Ivana Cibulkova.

**Writing – review & editing:** Veronika Rehorova, Hana Soukupova, Petr Waldauf, Monika Cahova, Jan Manak, Martin Matejovic, Frantisek Duska.

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
