## [Decision Letter · Decision Letter 0]

24 Jul 2024

PONE-D-24-03961Allogenic Faecal Microbiota Transplantation for Antibiotic-Associated Diarrhoea in Critically Ill Patients (FEBATRICE) – Study Protocol for a Multi-Centre Randomised Controlled Trial (Phase II).PLOS ONE

Dear Dr. Duška,

Thank you for submitting your manuscript to PLOS ONE. After careful consideration, we feel that it has merit but does not fully meet PLOS ONE’s publication criteria as it currently stands. Therefore, we invite you to submit a revised version of the manuscript that addresses the points raised during the review process.

Please go through the reviewers' comments carefully and address them accordingly.

We look forward to receiving your revised manuscript.

Kind regards,

Furqan Kabir

Academic Editor

PLOS ONE

4. One of the noted authors is a group or consortium [FEBATRICE (Faecal Bacteriotherapy in Intensive Care) Consortium  investigators ]. In addition to naming the author group, please list the individual authors and affiliations within this group in the acknowledgments section of your manuscript. Please also indicate clearly a lead author for this group along with a contact email address.

5. We note that the original protocol that you have uploaded as a Supporting Information file contains an institutional logo. As this logo is likely copyrighted, we ask that you please remove it from this file and upload an updated version upon resubmission.

Reviewers' comments:

Reviewer's Responses to Questions

**Comments to the Author**

1. Does the manuscript provide a valid rationale for the proposed study, with clearly identified and justified research questions?

Reviewer #1: Yes

Reviewer #2: Yes

2. Is the protocol technically sound and planned in a manner that will lead to a meaningful outcome and allow testing the stated hypotheses?

Reviewer #1: Partly

Reviewer #2: Yes

3. Is the methodology feasible and described in sufficient detail to allow the work to be replicable?

Reviewer #1: No

Reviewer #2: Yes

4. Have the authors described where all data underlying the findings will be made available when the study is complete?

Reviewer #1: Yes

Reviewer #2: Yes

5. Is the manuscript presented in an intelligible fashion and written in standard English?

Reviewer #1: No

Reviewer #2: Yes

6. Review Comments to the Author

You may also provide optional suggestions and comments to authors that they might find helpful in planning their study.

Reviewer #1: General comments:

I felt that, in large part, this manuscript contained the required parts of a study protocol outlined in the PLOS ONE guidelines (https://journals.plos.org/plosone/s/submission-guidelines#loc-study-protocols). There are a few specific incidences where I thought description of the methods was lacking and I have noted those in the specific comments.

My major concern was that I have some doubts about the choice of a time-to-event (TTE) analyses for an outcome where the follow up is only seven days. The time scale is never mentioned here. If it is days, then there may be many "tied" times which can be problematic for TTE analyses. Furthermore, the sample size appears to be based on the difference of two percentages and yet the primary analyses calls for a hazard ratio which requires a time-to-event analysis. The power analyses should match the methods used in the primary outcome analyses.

I then found it odd that for an outcome with a 28-day follow up period, an odds ratio is called for (line 379). Somewhere in that paragraph (lines 378-382) it should be mentioned that logistic regression is being used. Kaplan-Meier curves might be more useful for this longer follow up…but why run a non-TTE analysis and then a TTE analysis? I'm not following the logic here.

Finally, the majority of the protocol is written well enough, there are some instances of missing definite articles, singular forms used when plurals are correct, etc. The authors should note that PLOS ONE submissions are not copyedited before publication and manuscripts that don't meet the language standards can be rejected.

Specific comments:

1. (lines 154-155) This is not a complete sentence.

2. Per the PLOS ONE study protocol guidelines, the aim of the study should be stated in the Materials and Methods section. The primary and secondary outcome sections come close, but I'm not sure that they quite capture the aims.

3. (lines 316-323) Please reference the package(s) in R or the randomization algorithm used to implement the randomization.

4. (lines 383-391) It would be helpful to explain why each of these methods are being used. Some I am not familiar with, but others I know and am curious as to how you will use those methods.

5. (lines 408-409) The methods for stopping for futility should be articulated here or in supplementary materials.

Reviewer #2: Thank you for the opportunity to review the manuscript "Allogenic Faecal Microbiota Transplantation for Antibiotic-Associated Diarrhoea in Critically Ill Patients (FEBATRICE) - Study Protocol for a Multi-Centre Randomised Controlled Trial (Phase II)".

The manuscript is the protocol for a multicentre unblinded non-placebo-controlled phase III study to be carried out. I have minor questions only:

1/ What is HDU (line 163)?

2/ I suppose that "inc." (line 188) means "including", but "incl." (line 196) is also "including"?

3/ Line 203. Please correct "for one our" to "for one hour".

4/ Lines 207, 228, 301. Please correct "clostrioides" to "clostridioides"

7. PLOS authors have the option to publish the peer review history of their article (what does this mean?). If published, this will include your full peer review and any attached files.

Reviewer #1: No

Reviewer #2: **Yes: **Carlos Eduardo Pompilio

---

## [Author Response · Author response to Decision Letter 0]

30 Jul 2024

Please see detailed rebuttal letter below.

---

## [Editor Report · Decision Letter 1]

27 Aug 2024

Allogenic faecal microbiota transplantation for antibiotic-associated diarrhoea in critically ill patients (FEBATRICE) – Study protocol for a multi-centre randomised controlled trial (phase II).

PONE-D-24-03961R1

Dear Dr. Duška,

We’re pleased to inform you that your manuscript has been judged scientifically suitable for publication and will be formally accepted for publication once it meets all outstanding technical requirements.

Kind regards,

Furqan Kabir

Academic Editor

PLOS ONE
---

## [Editor Report · Acceptance letter]

10 Sep 2024

PONE-D-24-03961R1 

PLOS ONE

Dear Dr. Duska, 

I'm pleased to inform you that your manuscript has been deemed suitable for publication in PLOS ONE. Congratulations! Your manuscript is now being handed over to our production team.

Kind regards, 

on behalf of

Dr. Furqan Kabir 

Academic Editor

PLOS ONE